# Using Local Toponyms to Reconstruct the Historical River Networks in Hubei Province, China

**Aini Zhong [1,2], Yukun Wu [2], Ke Nie [1,3] and Mengjun Kang [1,2,*]**

[1]  Key Laboratory of Urban Land Resources Monitoring and Simulation, Ministry of Natural Resources, Shenzhen 518034, China; ainy_zhong@whu.edu.cn (A.Z.); nieke@whu.edu.cn (K.N.)

[2]  School of Resource and Environment Science, Wuhan University, No.129 Luoyu Road, Hongshan District, Wuhan 430079, China; wuyukun@whu.edu.cn

[3]  Shenzhen Research Center of Digital City Engineering, Shenzhen 518034, China

[*]  Correspondence: mengjunk@whu.edu.cn; Tel.: +86-1517-244-8589

**Abstract:** As an important data source for historical geography research, toponyms reflect the human activities and natural landscapes within a certain area and time period. In this paper, a novel quantitative method of reconstructing historical river networks using toponyms with the characteristics of water and direction is proposed. It is suitable for the study area which possesses rich water resources. To reconstruct the historical shape of the river network, (1) water-related toponyms and direction-related toponyms are extracted as two datasets based on the key words in each village toponym; (2) the feasibility of the river network reconstruction based on these toponyms is validated via a quantitative analysis, according to the spatial distributions of toponyms and rivers; (3) the reconstructed historical shape of the river network can be obtained via qualitative knowledge and geometrical analysis; and (4) the reconstructed rivers are visualized to display their general historical trends and shapes. The results of this paper demonstrate the global correlation and local differences between the toponyms and the river network. The historical river dynamics are revealed and can be proven by ancient maps and local chronicles. The proposed method provides a novel way to reconstruct historical river network shapes using toponym datasets.

**Keywords:** toponym quantitative analysis; historical landscape reconstruction; geographical weighted regression; Thiessen polygon; spatial cluster

## 1. Introduction

Researchers are often faced with a deficiency of data sources when investigating the historical evolution processes of various landscapes [1]. Archeological data and historical cartographic documents are traditional sources of geographic information regarding historical entities [2], but such data often appear to be subjective. In addition, it is difficult to discern the useful geographic information from these types of data without human recognition and interpretation. Alternatively, toponyms represent a powerful data source for historical geographical information [1].

A toponym is a general name for any place or geographical entity [3] that records the human activities and natural landscape characteristics around an area during a certain period, and thus, it can serve as a novel data source for historical geographical information [4]. In China, most places are named toponymically, and they are usually related to the landscape and location or to a wonderful moral, historical factor or a celebrity related to that place [1]. For example, the word "Hunan" of Hunan Province literally means that is located to the south of Dongting Lake; meanwhile, "Hubei" of Hubei Province means that it is located to the north of Dongting Lake. Toponyms, which serve as symbols of the regional culture, reveal the regional history, habitat and environment. Compared with

other geographical elements, toponyms are relatively conservative and less likely to change with time, and thus, the regional historical information may be preserved within them. Taking advantage of the spatial and temporal information implied within toponyms could assist with historical geographical research regarding a particular region or period when alternative data sources are absent. Furthermore, toponyms could play a significant role in the reconstruction of historical landscape characteristics.

Toponyms describe real existing waterbodies within a historical period based on human activities and terrain conditions. Although digital elevation models (DEMs) represent an effective method of delineating watercourses [5–7], this tool only identifies terrain characteristics and cannot reveal ancient features. In this paper, we study toponyms in Hubei province, China, where water resources are very abundant. Toponyms containing words related to water bodies are widespread throughout the study area [8], as are toponyms for localities that reference a direction. Using toponyms as the data source may provide some historical information about the river network. As we demonstrate in Section 5.2 of this paper, there is a strong correlation between river-related toponym datasets and river networks. Accordingly, with the help of direction-related toponyms, the historical shapes of river networks, particularly those in areas with numerous river-related toponyms, can be reconstructed using spatial analysis methods. In consideration of the historical changes in the channels of river networks evidenced by existing historical documents in Hubei Province, toponyms throughout the middle-lower Yangtze Plain, where the abovementioned correlation is stronger, may be affected by densely distributed rivers and lakes when employed to reveal the landscape dynamics. Therefore, the historical shape of the river network can be reconstructed using geometrical analysis methods based on a correlation analysis conducted to quantify the relationship between the toponym dataset and river density.

There are four sections in this paper. The first section introduces and summarizes previous research on toponyms, and gives an explanation of the significance of toponym-based landscape reconstruction efforts. The second section introduces the study area and outlines the methods adopted in this research for the data processing, correlation analysis and landscape reconstruction. The third section describes the results of the experiment. A toponym dataset is collected to serve as the data source, after which statistical information is calculated to confirm the global and local correlations between the toponyms and the river network. After finishing the reconstruction via geometric analysis combined with background knowledge, a comparison is performed between the reconstructed historical and present-day river shapes to determine the variation in the landscape. In this paper, we only reconstruct first- and second-level rivers since they are the main rivers with dynamics throughout the historical period. Finally, the fourth section concludes the experiment. A strong correlation is evident between the toponyms and river network in Hubei Province; consequently, it is believed that toponyms can serve as a source of data for determining the landscape dynamics. However, there is room for improvement in our experiment.

## 2. Related Research

In China, the study of toponyms, also known as toponymy, dates back to the first century AD; this research was mainly focused on the origins and nomenclature of toponyms [9,10]. Scientists around the globe have previously studied the historical, etymological, philological and semantic aspects of names primarily using a diachronic approach [11–13]. Traditional toponomastic analysis fundamentally investigates the semantic aspects of toponyms by relying more on qualitative and descriptive methods without any spatial analysis of numerous place names. For instance, M. Gelling investigated the history and origins of English place names, which illustrate the context of their locations [14]. Furthermore, McDavid explored the linguistic geographic information expressed within toponyms by studying documentaries [15].

Nevertheless, as computer technology continues to advance rapidly, additional quantitative approaches are being introduced into the toponomastic research field. At the beginning, an analysis is framed by a traditional paradigm; however, tools for mathematical statistics are often involved

in the classification of toponyms to show their proportions [16] and other nonspatial characteristics. For instance, Stephen C. Jett identified positional and directional linguistic elements in Navajo place names for the Canyon de Chelly system in Arizona, after which he divided them into different categories according to named natural features and conducted a statistical analysis to demonstrate the environmental characteristics recorded by the toponyms [2]. Although quantitative analysis methods are currently being applied to toponomastic research, the essential value of a place name, which provides spatial and temporal information about a particular area, is still unknown.

With the assistance of a geographical information system (GIS), researchers can attempt to visualize the spatial characteristics of toponyms while mining them for geographic information via spatial analysis [17–19]. An examination of the distribution patterns of regional toponyms suggested that place names are relevant to various types of geographical information and human activities [20–24]. Toponymy has also been applied to the field of biology; for example, John J. Cox and his colleagues assessed the biogeographic indexes of faunal place names in the United States, the results of which indicated that the spatial patterns of toponyms are indicators of a species' historical distribution [25].

Researchers proposed a question in the 1980s that has been commonly disputed: can toponyms be utilized as a data source for determining changes in the landscape [20]? For instance, some scholars quantified the variance within the landscapes and toponyms in a study area between several periods, after which they finally concluded that place names in natural areas characterized by substantial land changes indicate not only the landscape dynamics, but also how such changes are perceived [20]. Geographers then began to realize that toponyms could be regarded as clues to the historical and cultural heritage of particular areas. Some scientists then associated place names with biological and botanical fields to infer the distributions of past habitats [26]. Marco Conedera took advantage of reconstructed land use patterns and examined the "burn" in southern Switzerland based on toponyms [11]. Wei Luo and his team investigated the relationship between terrain characteristics and Tai toponyms and revealed that some toponyms express the regional terrain through cartography [27]. Jaime Fagúndez and Jesús Izco explored the spatial distribution of place names and their diversity pattern using the spatial cluster method to raise awareness on vegetation protection issues [28,29]. Qian and her team researched the spatial patterns of ethnic groups by considering the naming principles of toponyms in the Chinese language [30]. Although many researchers have integrated multidisciplinary approaches and concepts into their research to enrich the analysis of toponyms, the quantitative relationships between toponyms and landscapes are unknown.

Exploring the spatial patterns of place names could provide valuable information on the historical dynamics of landscapes [22,24]. Unfortunately, their applications in the fields of geography and historical landscape reconstruction remain scarce. In addition, many researchers have only focused on information regarding landscapes and linguistics; therefore, they have rarely identified the directional information that place names supply. In this paper, direction-related toponyms are of vital importance to the reconstruction of river shapes. Since the direction information implied within Chinese toponyms tends to reference landscapes, the direction-related toponyms in our study area are largely descriptive of the regional river network; thus, we can apply them to a reconstruction of the historical river shape. Spatial cluster analysis constitutes a powerful approach for identifying different distribution modes [29]; thus, we select water-related place names to generate continuous reconstructed lines utilizing this approach. Toponyms represent the settlers' perception of their surroundings in a historical context; therefore, they can serve as a powerful data source for historical river network reconstruction.

## 3. Study Area and Data Sources

Hubei Province is located in the central region of China between 106°12′E and 114°14′E, and it named because of its position to the north of Dongting Lake shown as Figure 1. The study area boasts various types of landscapes, including mountains, hills and plains. The northwest is slightly higher with ranges such as the Wudang Mountains, while the Jianghan Plain, which contains a dense network of rivers, is located in the central and southern parts of the study area. In addition, the middle and

lower reaches of the Yangtze River and its enormous tributaries, the largest of which is the Han River, are located within Hubei Province, and some abandoned river channels of the Yangtze River are situated in the south. The Yangtze River and the Han River converge at Wuhan, the capital city of Hubei. Since Hubei Province is full of lakes and rivers, changes to the river network therein occur frequently. It is believed that toponyms in this study area can reveal the geographical characteristics of the area due to the influence of the rich water system.

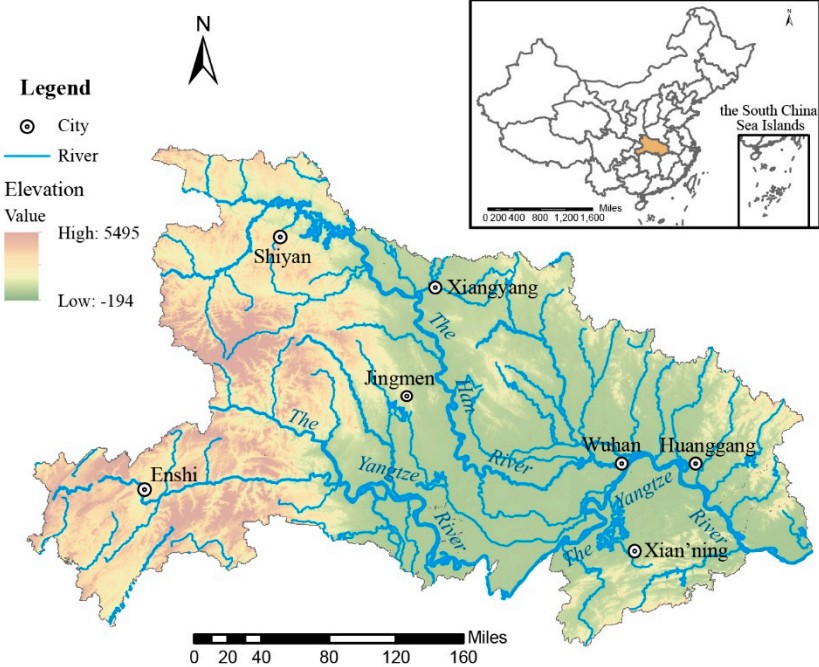

**Figure 1.** Location map of the study area.

To conduct this experiment, three categories of data, including datasets of the toponyms in the area, the current river network and historical documents, are collected. A total of 29,712 village toponyms are extracted, and their geometric center points are calculated as the locations of the corresponding toponyms. Village toponyms can serve as a reliable data source because they are less likely to change with time and are more conservative; moreover, they are more suitable for conducting spatial analysis than the toponyms of cities or counties. That is, they present fewer geometrical errors when taking their centroids as the locations of their toponyms. In addition, the river network within the study area, which refers to the Yangtze River (i.e., the longest river) and its numerous tributaries, is acquired from the Hubei Institute of Land Surveying and Mapping. Historical river network documents are collected from local chronicles of Hubei Province and the Changjiang Water Resources Commission of the Ministry of Water Resources.

## 4. Methods

The method used in this paper mainly consists of three parts. First, the toponyms are arranged and classified by screening key words in each one via manual judgement, after which two important datasets, namely, the datasets consisting of water-related toponyms and direction-related toponyms, are acquired. Second, a correlation analysis is performed between these two datasets and the river network before reconstructing the river network shape. The reconstruction results are meaningful only if there is a strong correlation. Finally, the reconstruction is conducted by using Thiessen polygons with a clustering algorithm in combination with curve smoothing to optimize the reconstruction of the historical river shape.

*4.1. Extraction of the Water-Related Toponym and Direction-Related Toponym Datasets*

A general mandarin toponym is primarily composed of a specific component and a generic component [31]. A generic name within a toponym usually functions as a qualitative summary of the common, essential features of the geographical object, while a specific name distinguishes different entities and is generally associated with the surrounding landscapes. Therefore, we can utilize this type of toponym structure to extract the water-related toponym and direction-related toponym datasets for the following analysis.

The water-related toponym dataset (WRTD) is composed of toponyms originating from vicinal waterbodies by human judgement. The toponyms containing water-related generic names listed in Table 1 are identified and extracted to the WRTD. These key words are derived from research on generic toponym names in our study area [8]. To ensure the integrity and unambiguity of the WRTD, the toponyms describing specific rivers are also added to the WRTD. For example, the place name "Baishi Village", which has no water-related generic names but refers to the Baishi River, should be collected in the WRTD (specific river toponyms are listed in Appendix A).

**Table 1.** Water-related key words in the generic names of Hubei Province.

| Generic Name | Example |
| --- | --- |
| Bin | Hubin Village |
| Cao | Changcao Village |
| Cha | Chahe Village |
| Chao | Gaochao Village |
| Chi | Meichi Village |
| Chuan | Gaochuan Village |
| Di | Yuedi Village |
| Du | Dadu Village |
| Fu | Fushui Village |
| Gang | Gangbian Village |
| Gou | Changgou Village |
| Hai | Honghai Village |
| He | Shahe Village |
| Hong | Hanhong Village |
| Hu | Donghu Village |
| Ji | Niji Village |
| Jian | Jianchi Village |
| Jiang | Bianjiang Village |
| Kou | Hekou Village |
| Liu | Heiliu Village |
| Qu | Changqu Village |
| Quan | Jinquan Village |
| Shui | Shuiping Village |
| Tan | Xiaotan Village |
| Tang | Shatang Village |
| Tuo | Liantuo Village |
| Wan | Wujia Wan Village |
| Xi | Lingxi Village |
| Yan | Shangyan Village |
| Yang | Baiyang Village |
| Yuan | Longyuan Village |
| Zhou | Longzhou Village |

The direction-related toponym dataset (DRTD) is a collection of toponyms that indicate locations by direction-related key words, such as "Nan" and "Bei", which denote south and north, respectively. Furthermore, the major locality toponyms in Hubei Province are based on the river network, which supports the relative positional relationship between the region and the natural landscape.

Accordingly, the toponyms in the DRTD are regarded as a significant data source for the following correlation analysis and river shape reconstruction.

Six categories of selected direction-related key words commonly used in Mandarin toponyms are employed to describe directions in this study as shown in Table 2. By filtering all toponyms based on their origin, only toponyms using these words to identify locations are extracted to form the DRTD. In addition, each direction-related toponym has an attribute $A_d$ representing its direction relative to a nearby river according to the origin and the regulation of naming for Mandarin place names.

**Table 2.** Selected key words for direction in specific names.

| Specific Name | Meaning | Example | $A_d$ |
|---|---|---|---|
| Dong | East | Hedong Village | E |
| Nan | South | Jiangnan Village | N |
| Xi | West | Huxi Village | W |
| Bei | North | Hongbei Village | N |
| Yin | Usually the north of the water | Xiangyinwan Village | N |
| Yang | Usually the south of the water | Chaoyang Village | S |

An administrative region is not suitable as the statistical unit because it often refers to the modifiable areal unit problem [32,33]. This problem reflects that statistical results vary at different scales, which is a common issue in quantitative studies regarding spatial phenomena. Otherwise, geographic phenomena are interrelated and contiguous at a spatial scale as mentioned in Tobler's First Law of Geography [34]. Therefore, we transform the boundary polygon of study area into a fishnet with 25 km × 25 km grids and only reserve grids within the boundary. Each grid serves as a statistical unit. Regular grids and constant area of units makes the result more similar to the geographical phenomenon of continuous distribution.

The density of water-related toponyms ($P_w$) and direction-related toponyms ($P_d$) in every statistical unit are calculated using Formula 1, where $P$ denotes the number of toponyms per unit area, $n$ is the number of water-related or direction-related toponyms in a statistical unit, and $A$ is the area of that unit:

$$P = \frac{n}{A} \tag{1}$$

*4.2. Correlation Analysis Between the River Network and Toponyms*

A correlation analysis is utilized as the foundation for reconstructing the river network. Ordinary Least Squares (OLS) is commonly used to verify the relationship between variables. The result shows a positive and weak correlation between $P_w$ and $D_w$ as well as between $P_d$ and $D_w$ after we apply OLS. Correlation coefficient $R$ between $P_w$ and $D_w$ is 0.293, while that of $P_d$ and $D_w$ is 0.374. However, OLS identifies the overall effects in a single model but does not account for the spatial dependence of neighboring units. To quantify the correlations between the toponym datasets and the river network, the geographically weighted regression (GWR) method is employed to measure the global association and spatial variations.

Each statistical unit $u_i$ contains three variables expressed as ($P_w$, $P_d$, $D_w$); the variable $D_w$ represents the river network density in a certain unit calculated using Formula 2, where $\sum L$ is the total length of rivers within $u_i$; and $A$ represents the area of $u_i$:

$$D_w = \frac{\sum L}{A} \tag{2}$$

The variable $D_w$ is chosen to reflect the regional hydrographic environment in each statistical unit; therefore, its correlation with the variables $P_w$ and $P_d$ in each county is discussed subsequently.

GWR is a tool for spatial-dependent variables to analyze the correlation among the local characteristics. It is a local form of linear regression used to model spatially varying relationships; consequently, we can explore the spatial heterogeneities of factors related to a certain phenomenon [35].

To quantify the spatial correlations between the toponym datasets and the river network, $P_w$ and $P_d$ are separately taken as consequent variables with $D_w$ as the argument. The correlations between $P_w$ and $D_w$ as well as $P_w$ and $D_w$ in each unit are analyzed using the GWR toolbox of ArcGIS developed by the Environment Sources Research Institution (www.esri.com) with an adapted kernel type and a corrected Akaike Information Criterion (AICc) bandwidth, the results of which can reveal global correlations by the adjusted $R^2$ and the local relevancy through a local $R^2$ value. In each unit, the local $R^2$ value between $P_w$ and $D_w$ ($LR^2{}_{wi}$) and that between $P_d$ and $D_w$ ($LR^2{}_{di}$) are calculated using Formulas (3) and (4), respectively, where $D_i$ is the real $D_w$ value of the unit $i$, $\hat{D}_{wi}$ and $\hat{D}_{di}$ are the $D_w$ values predicted according to the local GWR models, and $\overline{D}$ represents the mean value of all $D_w$.

$$LR^2{}_{wi} = 1 - \sum_{i=1}^{n} \frac{(D_i - \hat{D}_{wi})^2}{(D_i - \overline{D})^2} \tag{3}$$

$$LR^2{}_{di} = 1 - \sum_{i=1}^{n} \frac{(D_i - \hat{D}_{di})^2}{(D_i - \overline{D})^2} \tag{4}$$

The calculated $LR^2{}_{wi}$ and $LR^2{}_{di}$ values can reveal the goodness of fit of the local model in each statistical unit, and thus, the spatial disparity can be implied through choropleth maps of the local $R^2$ values (Section 5.3).

*4.3. Reconstruction of the Historical River Network*

In this section, we employ three steps to reconstruct the historical river network. First, Thiessen polygons are created according to the spatial pattern of toponyms in the DRTD. Second, cluster analysis is employed to distinguish local clusters and make the extracted lines more equivalent to the expected reconstructed result. Third, curve smoothing is utilized to simplify and optimize the reconstructed river shape.

4.3.1. Raw Reconstruction Results

Thiessen polygons define individual areas of influence around each set of points. Each Thiessen polygon contains a single input point feature. The polygons are used to divide an area covered by input points into several zones where any location within a zone is closer to its central input point than to any other input point.

Accordingly, if two input toponym points in adjacent Thiessen polygons refer to opposite directions, their common edge most likely reveals the real historical river orientation. The raw reconstruction result for the historical river network is achieved through the following steps:

1.　take representative points of the DRTD as the input to generate Thiessen polygons;
2.　turn the Thiessen polygons into lines while identifying the directions on either side, e.g., $A_d$(N, S);
3.　extract the edges that reflect opposite directions on either side with $A_d$(N, S) or $A_d$(E, W), as shown in Figure 2.

The points in Figure 2 are toponyms with their representative directions, the black lines are Thiessen polygon edges, and the red line represents the extracted line we desire.

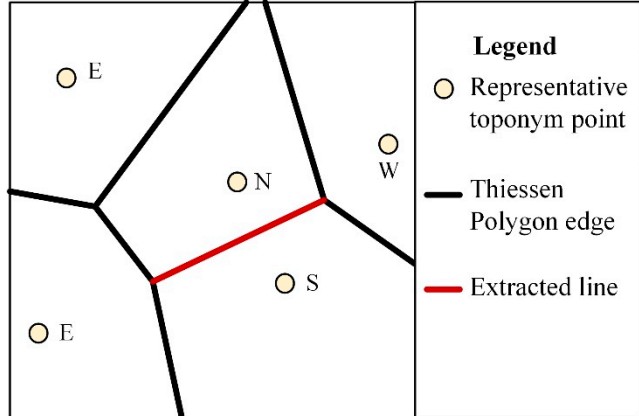

**Figure 2.** Sketched map of the extraction of Thiessen polygon edges.

### 4.3.2. Continuous Reconstruction Results

Cluster analysis represents the process of gathering a set of objects in a way that objects within the same group are more similar or spatially closer to one another than to those in other groups.

To obtain continuous reconstruction results, we conduct cluster analysis to eliminate the impacts of marginal effects on the reconstruction and form cluster groups of toponyms that possibly describe the same stream. Because the toponyms in the WRTD contain key words referring to different levels of rivers, their general shape can be better understood through a cluster analysis of different levels of rivers. Through the clustering of different specific water-related toponyms, this step helps the reconstruction move forward and makes it easier for us to draw the orientations of historical rivers. The steps are as follows:

1.  classify toponyms in the WRTD into different categories according to the levels of rivers they refer to, where $L_i$ represents each category;
2.  take the representative points of $L_i$ as the input to conduct hierarchical clustering in SPSS on the basis of the Euclidean distances between those points;
3.  determine an appropriate number of clusters as $N_c$ according to the tree map from the previous step;
4.  conduct the group clustering of $N_c$ using the ArcGIS toolbox to obtain clusters;
5.  add layers of grouping analysis on $L_i$ to the raw reconstruction results, after which a continuous historical river skeleton at each level can be drawn from the overlay based on trend identification and qualitative knowledge. The connection is completed with some algorithm and manual supervision, according to following principles: (1) Raw reconstruction results within two standard deviational ellipse polygons of clustering result should be extracted as baseline. (2) Search nearest neighbor points around line segments from clustering result, then connect both ends with each nearest neighbor points and repeat this step to form continuous feature. (3) Make sure the local direction of connection is consistent with manually identified trend about clustering result.

### 4.3.3. River Shape Optimization

Due to the lack of adequate nodes within the extracted lines, the results exhibit geometrical zigzag features after extracting the Thiessen polygons and performing the cluster analysis. To dispose of acute angles between the zigzagging lines, curve smoothing is adopted to smooth the lines, thereby optimizing the results. A Bezier interpolation method improves the aesthetic or cartographic qualities of the lines by creating Bezier curves to fit and go through each input line.

## 5. Results and Discussion

### 5.1. Results of the WRTD and DRTD Extraction

After collection, 5234 toponyms are included in the WRTD, which shows that the toponyms in Hubei Province are rich in water-related place names because they account for 19.15% of all toponyms. The statistical data regarding the spatial distribution and proportions of toponyms in the WRTD are displayed in Table 3. Most toponyms in the WRDT are within the 10 km river buffer, and a greater number of toponyms are situated close to rivers, indicating that river networks have a greater impact on the naming of closer place, which is supported by the quantity variance between these buffers. This information indicates the strong dependence of many place name definitions on the distribution of the river network in the study area due to the high coverage of natural water resources.

**Table 3.** Proportions of toponyms in the water-related toponym dataset (WRTD) in different buffers.

| Buffer radius (km) | Number | Proportion (%) |
|:---:|:---:|:---:|
| 1 | 924 | 17.65 |
| 3 | 2071 | 39.57 |
| 5 | 2826 | 53.99 |
| 8 | 3707 | 70.83 |
| 10 | 4159 | 79.46 |

The total number of toponyms in the DRTD is 1575. Table 4 lists information about the proportions of toponyms in the DRTD in different buffers after we employ the buffer analysis. The results indicate that most toponyms in the DRTD are located near river networks and $A_d$ of them are greatly based on nearby rivers, which is similar to the results of the WRTD.

**Table 4.** Proportions of toponyms in the direction-related toponym dataset (DRTD) in different buffers.

| Buffer Radius (km) | Number | Proportion (%) |
|:---:|:---:|:---:|
| 1 | 247 | 15.68 |
| 3 | 594 | 37.71 |
| 5 | 842 | 53.46 |
| 8 | 1086 | 68.95 |
| 10 | 1205 | 76.51 |

### 5.2. Variables of Statistical Units

Taking 25 km × 25 km grids as statistical units, the variables ($P_w$, $P_d$, and $D_w$) of a total of 227 units are analyzed from a spatial perspective.

The river network density $D_w$ is employed in every unit as the natural index of the river network. The river networks identified by each unit are assigned an index $D_w$ following Formula 2. The spatial pattern of the river network over the study area is shown through the choropleth map in Figure 3a, which indicates that the regions with dense river networks are mainly concentrated in the central and southeastern parts of the study area, with a smaller number in the northern part.

Acting as regional indexes of water-related toponyms and direction-related toponyms, the $P_w$ and $P_d$ of each unit are calculated according to Formula 1 as mentioned above. The spatial pattern of the $P_w$ and $P_d$ values is generally consistent with the river distribution as shown in Figure 3b,c. The grids with a high value of $P_w$ and $P_d$ cluster around the southeastern part of the study area, and the distribution of $P_w$ values is more consistent with the river network density. Thus, grids of dense river networks appear to contain dense water-related and direction-related toponyms as well.

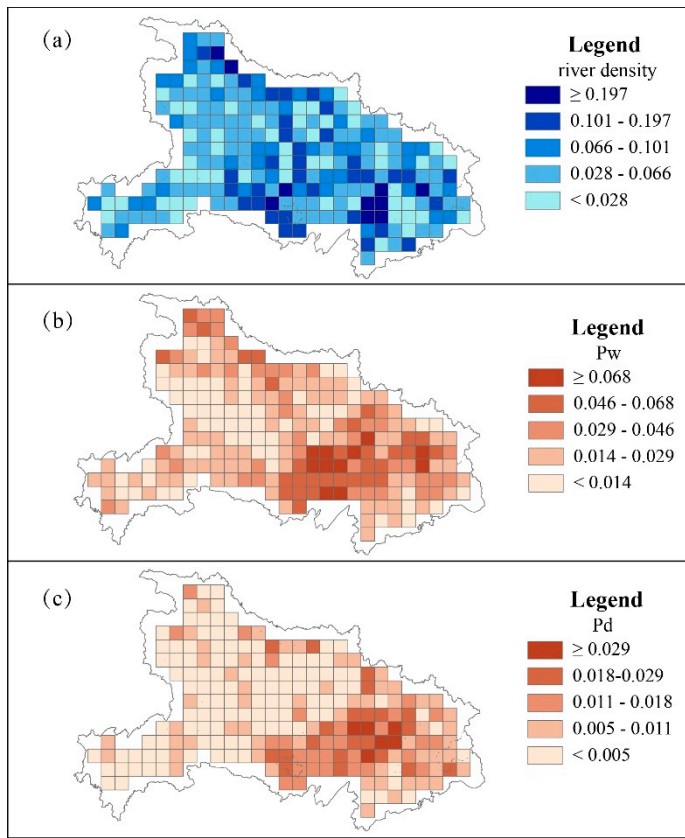

**Figure 3.** Choropleth maps of $P_w$, $P_d$, and $D_w$ at the unit level, (**a**) Choropleth map of the river network density at unit level; (**b**) Choropleth maps of $P_w$ at the unit level; (**c**) Choropleth maps of $P_d$ at the unit level.

Accordingly, evidence of a strong association between the naming of toponyms and the river networks in Hubei Province can be provided by correlation analyses between $P_w$ and $D_w$ and between $P_d$ and $D_w$.

### 5.3. Correlation Analysis

To reveal the statistical disparity between the two pairs of variables (i.e., $P_w$ and $D_w$ and $P_d$ and $D_w$), the mean value, standard deviation, maximum value and minimum value are calculated and listed in Table 5.

**Table 5.** Statistical features of the variables.

| Variables | Mean Value | Standard Deviation | Max Value | Minimum Value |
|:---:|:---:|:---:|:---:|:---:|
| $D_w$ | 0.094 | 0.077 | 0.387 | 0.00 |
| $P_w$ | 0.189 | 0.071 | 0.500 | 0.00 |
| $P_d$ | 0.054 | 0.036 | 0.182 | 0.00 |

To study the spatial disparity in the relevancy between $P_w$ and $D_w$ and $P_d$ and $D_w$ and validate the feasibility of our reconstruction, the GWR method is performed to analyze the global and local relationships. In the social sciences, correlation coefficient values over 0.6 represent a strong correlation between two variables, and a strong relationship is indicated by $R^2$ values over 0.36 [36]. Taking $P_w$ as the consequent variable and $D_w$ as the argument, the residual square of the model is 2.063 and the global adjusted $R^2$ value is 0.620; thus, these findings reveal that the observed data present a good overall fit with this model. For $P_d$ and $D_w$, the model also reveals a strong correlation, with a residual square of 0.005 and a global adjusted $R^2$ value of 0.576.

To discuss the local correlation in each unit, Choropleth maps are shown in Figure 4 to visualize the local $R^2$ distribution by dividing the local model fit into several levels to highlight counties with high local $R^2$ values. Both maps demonstrate a spatial variation in the degree of model fit between the predicted and true values. The central to southern part of the study area exhibits a good fit since the dense river network greatly influences the spatial distribution characteristics of water-related and direction-related toponyms; that is, a denser river network in a particular region will have a more influential effect on the associated place names. Therefore, we can safely promote the opinion that toponyms are strongly affected by the presence of a rich river network and the water environment is recorded in the nomenclature throughout Hubei Province, especially the central part.

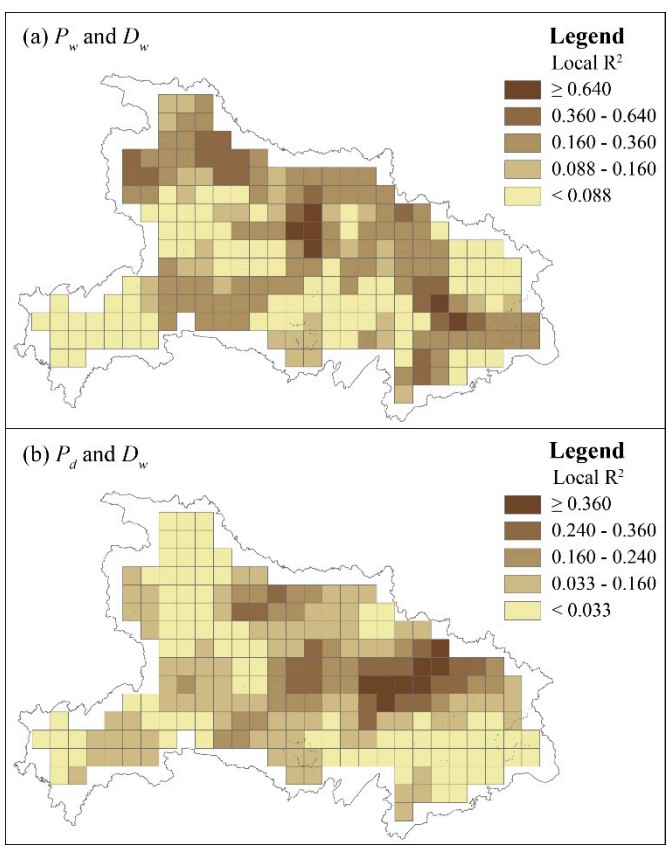

**Figure 4.** Choropleth maps of the local $R^2$ distribution at the unit level, (**a**) $P_w$ and $D_w$; (**b**) $P_d$ and $D_w$.

### 5.4. Reconstruction of the Historical River Network

Taking advantage of the properties of Thiessen polygon edges, we can extract the common lines between water-related toponyms that reference opposite directions, and thus, we can proceed to a reconstruction of the historical river shape. Thiessen polygons are generated according to the positions of the toponyms in the DRTD, after which they are converted from polygon boundaries to lines. Those lines within Hubei Province are clipped and reserved, the results of which are shown in Figure 5a. The spatial pattern of toponyms in the DRTD reveals that they are more densely gathered in the eastern part of the study area and more dispersed in the west; that is, the results of extracting lines from the Thiessen polygons appear similar to the spatial distribution of the polygons. In the south-central and southeastern parts of the study area, we can observe dense and clustered lines extracted from a dense distribution of relatively small Thiessen polygons with more edges; meanwhile, the lines in the western parts exhibit contrasting features. Figure 5b shows the spatial distribution of lines extracted from the Thiessen polygon edges.

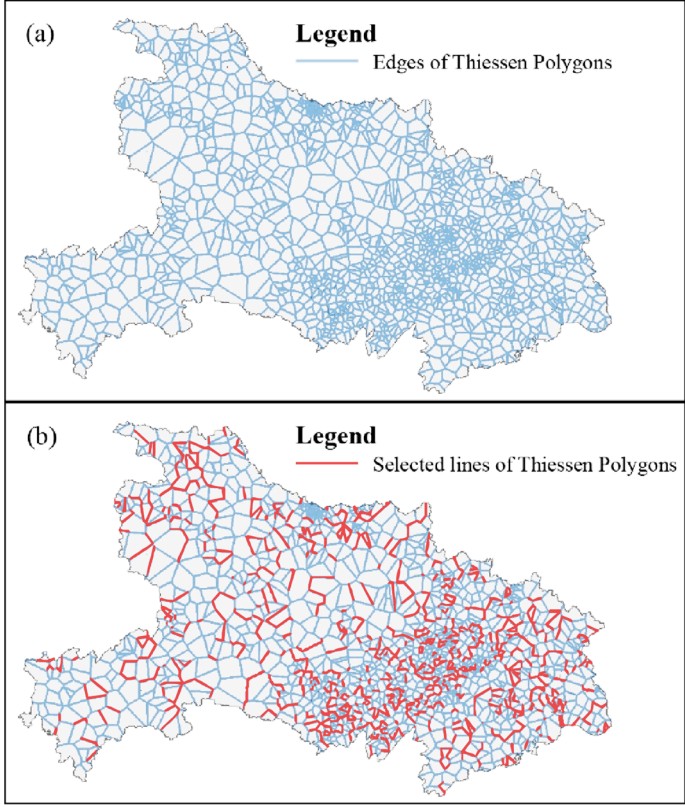

**Figure 5.** Unextracted and extracted lines from the Thiessen polygon edges, (**a**) unextracted lines from the Thiessen polygon edges; (**b**) extracted lines from the Thiessen polygon edges.

The lines extracted from the Thiessen polygon edges (Figure 5b) exhibit a similar spatial density as the river networks, but still present a disorderly distribution over the whole area, which obscures the true river orientation. To reconstruct the specific historical river shape, an optimization approach should be utilized to eliminate redundancy within the results. We obtain an appropriate number of clusters by performing hierarchical clustering on the basis of the Euclidean distances between water-related toponyms referring to particular river levels. In our experiment, we try to reconstruct the shapes of the main rivers (i.e., the Yangtze River and the Han River) in Hubei Province by water-related toponyms incorporating the word "Jiang", which definitely represents a first- or second-level river in Chinese. The toponyms of the WRDT containing this key word are selected as inputs for the hierarchical clustering, and the tree diagram of the analysis is shown in Figure 6. According to the tree diagram and to show the general trend, we first group the toponyms into 7 clusters via a clustering analysis based on spatial Delaunay triangulation constraints. As shown in Figure 7a, those 7 groups of clusters can be divided into several parts to show parts of the river orientations within the clusters. Subsequently, the toponyms are grouped into 12 parts to reveal more details of the river shape in Figure 7b.

Adding layers from the grouping analysis for "Jiang" to the extracted results in Figure 5b, we can draw continuous first- and second-level historical river skeletons from the overlay. When generating continuous reconstruction results, we only utilize extracted lines of Figure 5b within two standard deviational ellipse polygons (describe the spatial characteristics of geographic features: central tendency, dispersion, and directional trends) of the above clusters to ensure that these lines definitely indicate the orientation of our target rivers. Due to the terrain characteristics in China, where the land is topographically high in the west and low in the east, the rivers and streams mostly run from the west to the east. Therefore, the junction between the toponyms containing "Jiang" and the extracted results of Figure 5b should be operated according to the order of abscissa values in addition to the spatial continuity. Lines closer to toponyms with the key word "Jiang" are preferentially connected to

form the reconstructed result. Combined with manual identification and the application of qualitative knowledge, the reconstruction results can be modified to avoid fragmented rivers.

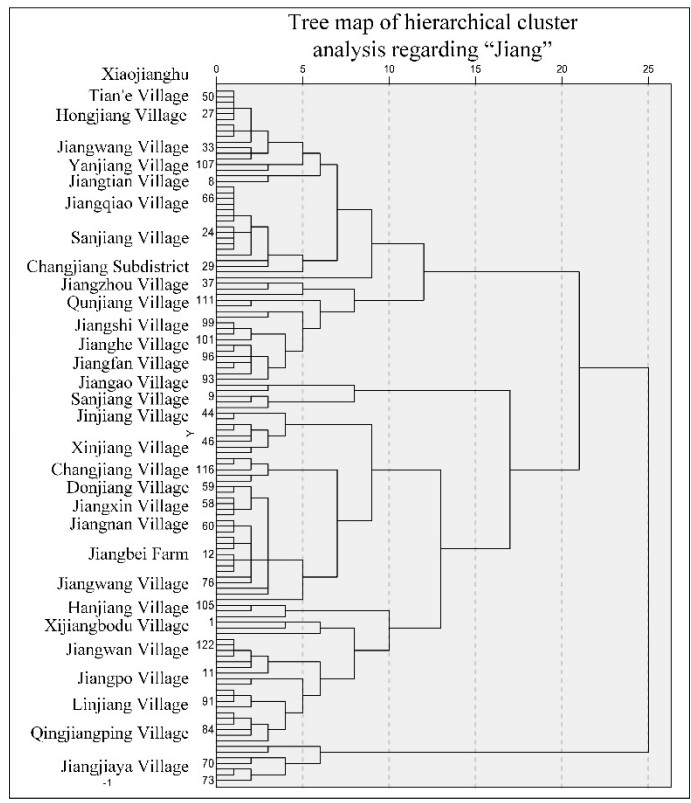

**Figure 6.** Tree diagram of hierarchical clustering for "Jiang".

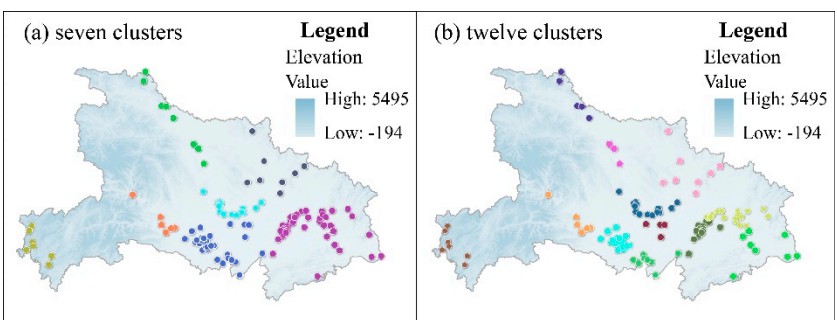

**Figure 7.** Cluster analysis of toponyms containing "Jiang", (**a**) seven clusters; (**b**) twelve clusters.

The final reconstruction result for the historical river shape is acquired after applying a curve smoothing algorithm to the lines handled through the abovementioned methods to smooth the original curves and eliminate noise effects. Figure 8 shows the reconstructed first- and second-level rivers. Some of the reconstruction results for the Han River deviate from the known path of the river. For instance, in boxes A and B, the reconstructed Han River exhibits more zigzag features; moreover, in box C, two lines intersect the Yangtze River. However, the reconstructed shape of the Yangtze River is almost identical to the present-day path, and even the geometrical characteristics at the county level are restored in spite of slight differences in box D. In addition, some redundant lines cover lands that do not currently possess first- or second-level rivers. Taking a holistic view of the results in Figure 8, the reconstruction of the river shape in the south-central and southeastern parts of the study area overall exhibit a good quality, and they are coincident with the present-day river pathways where there are numerous, densely concentrated water-related and direction-related toponyms. Meanwhile,

there are many redundant lines in the southwest and northeast. Considering the correlation analysis results in Figure 4, the reconstructed results around the central and southern part of the study area are more credible.

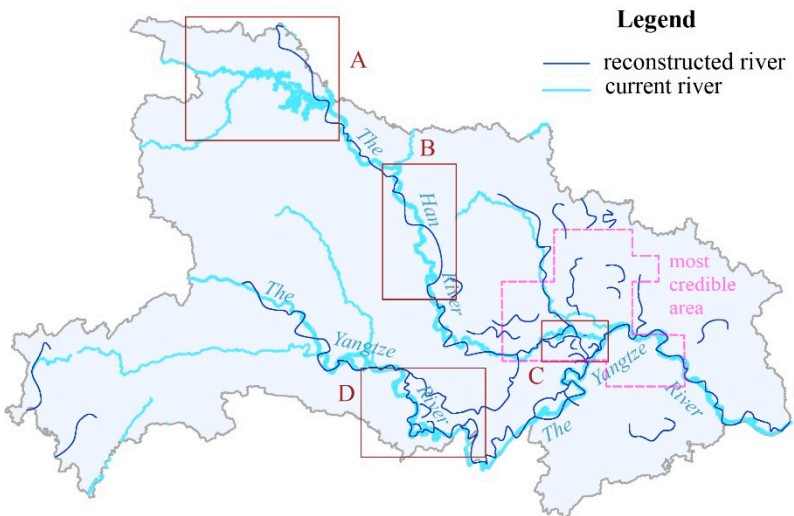

**Figure 8.** Reconstructed results for first- and second-level rivers.

The misfits of the river shapes in boxes A, B, C and D in Figure 8 may be caused by two reasons summarized below.

1.  No adequate toponyms:

The lines in boxes A and B zigzag with many angularities despite the curve smoothing process. These differences could result from an insufficient number and sparse distribution of toponyms for integrating the distribution of lines extracted from the Thiessen polygons with the grouping analysis results shown in Figure 7. Therefore, these angular lines are not the result of landscape changes. Moreover, there are no records about such events in the local chronicles of these regions. These differences should be regard as dynamics with low credibility, and only provide possible overall trends. Additionally, topographic and geomorphic features are often prominent in regions with sparse toponyms. Historical changes in that place can be extracted from DEM analysis as a detailed supplement to solve the problem, or more on historical records.

2.  River dynamics:

With the dense distribution of toponyms in these areas, the misfits in boxes C and D are considered the results of river dynamics, which can also be confirmed by historical maps and local chronicles. With regard to the Han River, box C demonstrates a change in the location where it enters the Yangtze River. Although this location is under dispute, the map of the river course in Figure 9 changes around the Han River [37] and shows the same river orientation as our reconstructed result in this region. Moreover, the misfit in box C can also be linked to river network changes according to an ancient book, namely, the History of Ming. Box D shows some lines of the Yangtze River that are generated by a natural curve cut-off phenomenon that forms oxbow lakes. The evidence of these dynamics can be observed in remote sensing images of Shishou and Jianli County, where many oxbow lakes are present.

To explore and verify the river dynamics around box D in Figure 8, we digitize ancient maps from different dynasties collected in The Historical Atlas of China [38] and map the main rivers belonging to the current Hubei Province. A comparison of the reconstructed rivers with the historical rivers in Figure 10 shows that the river dynamics in different times are integrated into the reconstructed results. The spatial–temporal characteristics of toponyms are recorders of landscapes, with each of them describing an event of a certain period, and overall, they indicate historical dynamics.

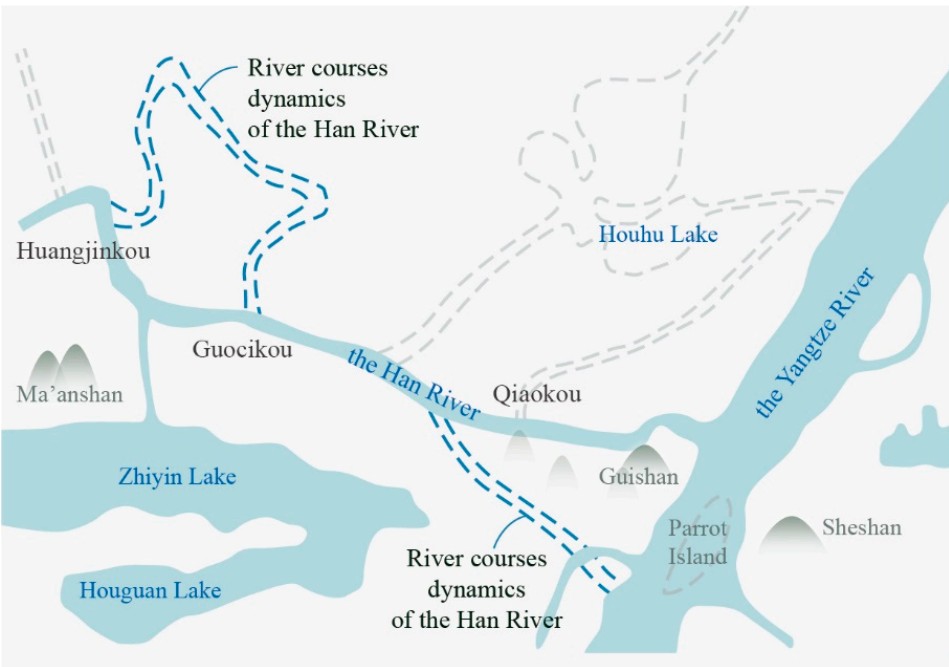

**Figure 9.** Map of the river course dynamics around the Han River.

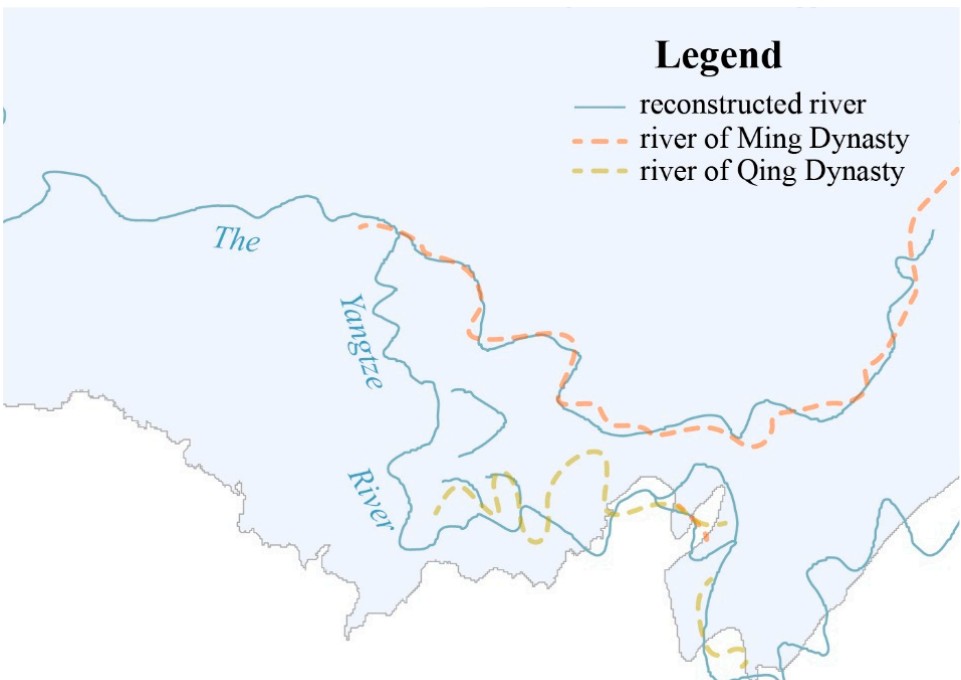

**Figure 10.** Map of river course dynamics about the Yangtze River.

As for the redundant lines and geometrical angular offsets, they may be caused by the extracted lines of the Thiessen polygons. The generated Thiessen polygons rely substantially on the distribution pattern of all toponyms in the DRTD; thus, an uneven distribution of place names referencing different locations can lead to varying densities of Thiessen polygons. The lines extracted from the Thiessen polygons will be more detailed and have fewer angular offsets with a denser concentration of direction-related toponyms in the study area; thus, the result is more likely to approach the real river shape. The extracted lines are used to reveal possible river orientations around our target rivers; therefore, lines describing other rivers are also integrated in the reconstructed results, which inevitably

leads to redundant lines. Meanwhile, the extraction of lines from pairs of Thiessen polygons requires not only a balanced spatial distribution of place names representing opposite directions but also similar counts in different directions. Although there are toponyms representing every possible direction in our study area, this is not the case for some marginal units. For instance, some direction-related place names may be concentrated in one place; alternatively, they may not be distributed along the entire trend of the described river or may be situated on only one side, resulting in discontinuous and scattered extraction results.

Our analysis is atemporal, because toponyms we based on are current data have no additional temporal information to refer to a specific period, but do preserve changes of the past. In this case, the result shows river network dynamics for all time periods. If this method is applied to toponyms of the same period, changes before this time can be extracted.

## 6. Conclusions

In this paper, we selected toponyms in Hubei Province as a sample of historical features to study the historical shape of the river network and study the landscape dynamics. Spatial statistical analysis was adopted to quantify the global and local correlations between the water- and direction-related toponyms and the river density. We also took advantage of geometrical analysis and spatial clustering to construct and optimize the historical shape of the river network while associating it with qualitative knowledge. Consequently, the experiment was able to demonstrate the historical river dynamics in Hubei Province. Based on the method proposed in this paper, the reconstructed shape of the river network is roughly consistent with the present-day network in the southeastern and south-central parts of the study, and the reconstruction results can partly reflect the historical river dynamics recorded in the ancient literature. This method of obtaining historical river shapes can be applied to ancient river dynamic research and is also helpful for mapping ancient rivers at a large scale.

However, there are some shortcomings in the proposed methodology that must be improved. The method of this paper is greatly influenced by the extracted and distribution characteristics of the toponyms in the region. In addition, we have not conducted a specific temporal analysis since we ignored the naming of certain toponyms, which may also affect the reconstruction result. In future research, toponyms of different dynasties will be collected to associate their generation and disappearance with river dynamics. Moreover, natural terrain characteristics should be taken into consideration to identify changes in terrain via the toponym analysis, which will lead to more effective results.

**Author Contributions:** Conceptualization, Ke Nie and Mengjun Kang; Data curation, Aini Zhong and Yukun Wu; Formal analysis, Aini Zhong and Yukun Wu; Funding acquisition, Ke Nie; Investigation, Aini Zhong and Yukun Wu; Methodology, Aini Zhong and Mengjun Kang; Resources, Ke Nie; Visualization, Aini Zhong; Writing—original draft, Aini Zhong; Writing—review & editing, Mengjun Kang. All authors have read and agreed to the published version of the manuscript.

**Funding:** The Project Supported by the Open Fund of Key Laboratory of Urban Land Resources Monitoring and Simulation, Ministry of Natural Resources, and the National Key Research and Development Program of China, grant number 2017YFB0503500.

**Acknowledgments:** This work was supported by the Open Fund of Key Laboratory of Urban Land Resources Monitoring and Simulation, Ministry of Natural Resources, and the National Key Research and Development Program of China (2017YFB0503500).

**Conflicts of Interest:** The authors declare no conflict of interest.

## Appendix A

Specific river toponyms adopted in Section 4.1 are listed here as supplementary data for detail information.

**Table A1.** Specific river toponyms.

| River | Toponym |
| --- | --- |
| The Baishi River | Baishi Village |
|  | Baishi Ping Village |
| The Shigu River | Shigu Village |
| The Qi River | Qilin Village |
| The Nie River | Niexing Village |
|  | Jingan Qiao Village |
|  | Dongjing Village |
| The Dongjing River | Zijing Village |
|  | Huangjing Village |
|  | Dongjing Forest Farm |
|  | Jingzhong Village |
|  | Jinghua Village |
|  | Jingfeng Village |
|  | Jingnan Village |
|  | Jinggan Village |
|  | Jinglan Village |
| The Jing River | Jingtan Village |
|  | Jingghong Village |
|  | Jingshi Village |
|  | Jingsha Village |
|  | Jingdong Village |
|  | Jingsong Village |
|  | Jingbei Village |
|  | Jingan Village |
| The Qing River | Qingyuan Village |
|  | Qingping Village |
|  | Yitian Men Village |
| The Tian River | Ertian Men Village |
|  | Santian Men Village |
| The Mian River | Mianyang Village |
|  | Miancheng Village |

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
