# Peer review of "Using Local Toponyms to Reconstruct the Historical River Networks in Hubei Province, China"

_ijgi, doi:10.3390/ijgi9050318_

Round 1

Reviewer 1 Report

An interesting paper that uses water-related and direction-related elements of toponyms to reconstruct historical river networks in Hubei province.  The method used is clearly outlined, however a little more detail is required on how line segments are combined to create contiguous features (see below).  The results are promising - in particular, the strong correlation between the reconstructed river and the course of the Yangtze during the Ming Dynasty shown in Figure 10.  The method and results presented in the paper appear to suggest that the analysis was atemporal, i.e. based on toponyms for all time periods.  This needs to be clarified and, if this is the case, the authors should comment on how effectively their methodology could be applied to subsets of toponyms that can be related to specific time periods.  The following minor points should also be addressed:

Lines 55-56 - suggest changing "From our perspective,..." to "As this paper demonstrates,..." and adding a cross reference to Section 5.2

Line 151 - add separator to the number of toponyms, i.e. 29,712

Lines 182-185 - add a list of specific river toponyms as an appendix

Line 199-205 - the modifiable areal unit problem applies to any dataset that is aggregated by areal unit, whether it is aggregated by administrative region or grid square.  The advantage of using grid squares in this instance is that the area of the units is constant.  This paragraph needs to be revise to reflect this.

Lines 212-214 - report test statistics

Line 236 - replace "fitness' with "goodness of fit"

Line 237 - add cross references to the test statistics reported in Section 5.3 and Figure 4

Lines 277-279 - clarify how the line segments for clusters are joined to create contiguous features.  Is this done manually or algorithmically? A little bit more detail - or perhaps a figure showing before/after - is required as this step is key to reconstructing historical river networks

Lines 331-332 - add citation

Figures 5a and 5b - change the symbology on this figure to either match the colour scheme on Figure 2 OR to use different colours for the polygon edges shown in figure 5a and the extracted edges shown in figure 5b.

Reviewer 2 Report

The paper provides a very interesting analysis on toponymy and spatial information in a specific location in China, based on transdisciplinar research. The background and the assumptions are well documented, despite some minor changes needed to strengthen the paper which are addressed below. Overall, the spatial analysis is also well documented and follows the assumptions of spatial models.

Line 32, 34, 41 108, 331/332: References are needed here

Line 55: Up to here readers are not introduced to the study area yet, despite the title and abstract. So I suggest to rewrite this sentence.

Line 66: There are four sections in this paper. "remaining" is a completely useless word here. Also, the first section is meant to be the introduction and not the background.

Could the authors provide some more comparision with other studies in their discussion, mainy from line 412 on?

Author Response

This manuscript is a resubmission of an earlier submission. The following is a list of the peer review reports and author responses from that submission.